# Immune profiling of age and adjuvant-specific activation of human blood mononuclear cells in vitro
Simone S. Schüller[1,2,7,9], Soumik Barman [1,2,9], Raul Mendez-Giraldez [3], Dheeraj Soni [1,2,8], John Daley[4], Lindsey R. Baden[2,5], Ofer Levy [1,2,6,10] ✉ & David J. Dowling [1,2,10] ✉

Vaccination reduces morbidity and mortality due to infections, but efficacy may be limited due to distinct immunogenicity at the extremes of age. This raises the possibility of employing adjuvants to enhance immunogenicity and protection. Early IFNγ production is a hallmark of effective vaccine immunogenicity in adults serving as a biomarker that may predict effective adjuvanticity. We utilized mass cytometry (CyTOF) to dissect the source of adjuvant-induced cytokine production in human blood mononuclear cells (BMCs) from newborns (~39-week-gestation), adults (~18-63 years old) and elders (>65 years of age) after stimulation with pattern recognition receptors agonist (PRRa) adjuvants. Dimensionality reduction analysis of CyTOF data mapped the BMC compartment, elucidated age-specific immune responses and profiled PRR-mediated activation of monocytes and DCs upon adjuvant stimulation. Furthermore, we demonstrated PRRa adjuvants mediated innate IFNγ induction and mapped NK cells as the key source of TLR7/8 agonist (TLR7/8a) specific innate IFNγ responses. Hierarchical clustering analysis revealed age and TLR7/8a-specific accumulation of innate IFNγ producing γδ T cells. Our study demonstrates the application of mass cytometry and cutting-edge computational approaches to characterize immune responses across immunologically distinct age groups and may inform identification of the bespoke adjuvantation systems tailored to enhance immunity in distinct vulnerable populations.

Distinct age-specific immune responses pose a challenge in combating diseases in early, as well as later in life. As such, it is important to understand how age-specific immune responses affect vaccine efficacy[1,2]. Various cell lineages from the innate and adaptive arm of the immune system are involved in triggering age-specific immune responses that ultimately induce immunological protection. Traditional vaccine development has largely overlooked distinct immunity in vulnerable populations, including newborns (≤ 28 days of life) and older adults (≥65 years of age)[3,4]. Important lessons could be learned by mapping the age-related changes in cell lineages involved in the innate and adaptive arms of the host immune responses.

Newborns have a distinct immune system, characterized by reduced inflammation and Th polarizing innate immunity and a bias towards Th2/Th17 responses[3,5,6], while elder individuals (>65 years of age) demonstrate immunosenescence, an age-related decline in general immune function,

with a decrease in naïve T cell production and an increase in memory T cells in tissues[7,8], resulting in a decreased ability to mount an effective immune response to new pathogens. Newborns are generally more dependent on their innate immune system for protection against infections[9]. CXCL8-mediated enhanced IFNγ expression in γδ T cells might be the important producers of effector cytokines in newborns, which are rare in adults[6]. TLR agonists can bypass neonatal Th2 biasedness[3,10,11] but might contribute to inflammaging in elders by inducing chronic inflammation via the production of pro-inflammatory cytokines[12]. TLR agonists have been used in vaccine development to help stimulate the immune system, but the selection of such an adjuvant should be ontogeny-specific to avoid inflammaging in elders[13]. Notably, innate IFN responses to TLR agonist adjuvant stimulation have been correlated with increased vaccine immunogenicity at the extremes of age[10,11,14–16].

[1]Precision Vaccines Program, Boston Children's Hospital, Boston, MA, USA. [2]Harvard Medical School, Boston, MA, USA. [3]Beckman Coulter Life Sciences, Brea, CA, USA. [4]Dana Farber CyTOF Core Facility, Dana-Farber Cancer Institute, Boston, MA, USA. [5]Department of Medicine, Brigham and Women's Hospital, Boston, MA, USA. [6]Broad Institute of MIT & Harvard, Cambridge, MA, USA. [7]Present address: Neonatal Directorate, Child and Adolescent Health Service, Perth, Australia.[8]Present address: Sanofi, Cambridge, MA, USA. [9]These authors contributed equally: Simone S. Schüller, Soumik Barman. [10]These authors jointly supervised this work: Ofer Levy, David J. Dowling. ✉e-mail: ofer.levy@childrens.harvard.edu; david.dowling@childrens.harvard.edu

Newborns have a lower proportion of γδ T cells and invariant Natural Killer T (iNKT) cells, but similar frequency of NK cells compared to adults[17]. Both frequency and number of γδ T cells rise in adulthood and decline in aged individuals[18]. Monocytes from elder individuals exhibit a reduced ability to produce cytokines in response to PRRa stimulation compared to monocytes from younger individuals[19]. NK cells from elder individuals also show reduced cytotoxicity and cytokine production compared to NK cells from younger individuals[20]. Elder individuals have a decreased number of mature B cells compared to adults[21], which is a direct consequence of impaired adaptive immunity in the aged setting. Such age-related changes in the innate/adaptive immune cell compartment and production of cytokines upon PRRa adjuvanted vaccination, which shape the adaptive immune responses could be easily learned by immunophenotyping.

Immunophenotyping is a technique used to identify and characterize different types of immune cells in a sample, based on their surface markers or antigens. Immunophenotyping could be achievable by multiparameter flow cytometry or by mass cytometry (CyTOF). CyTOF is a powerful technology that allows researchers to analyze immune cells in greater detail than traditional flow cytometry[22–24]. By analyzing multiple parameters at once (i.e., >50) regarding the protein expression of the cells, different types of immune cells in a particular sample could be easily identified and characterized. During the COVID-19 pandemic, CyTOF was extensively used to study the immune response to the virus, and to identify potential biomarkers[25–28] or targets for new treatments or therapies[29–31]. Overall, the success of CyTOF during the last decade highlights the importance of advanced technologies in characterizing the immune response to pathogens, and in developing new treatments or therapies to combat the disease[32–34].

High-dimensional CyTOF data is complex and requires sophisticated computational analysis to extract meaningful biological insights. Machine learning (ML) algorithms have the potential to revolutionize high-dimensional flow cytometry (conventional or spectral) and CyTOF data analysis[35]. ML algorithms can be used for cell population identification, classification, and prediction, which can save time and reduce human error[35]. Moreover, deep learning algorithms can learn complex patterns in high-dimensional data, enabling the discovery of novel cell subsets or biomarkers. One of the various approaches for analyzing high-dimensional data is a cloud-based software, the Cytobank platform, which offers dimensionality reduction (DR), clustering, and differential expression analysis[36–38]. However, careful consideration of data preprocessing, normalization, and quality control is essential to ensure accurate and reproducible results[39,40].

Generally, vaccine responses are elicited by a variety of platforms that include inactivated, live attenuated, toxoid, subunit, recombinant, polysaccharide, and conjugate vaccines with adjuvants. Up until the early 21st century, alum remained the only adjuvant included in licensed vaccine formulations, until, in 2015 MF59 (squalene emulsion) was incorporated into a licensed influenza vaccine (Trade name: *Fluad*) designed for enhancing efficacy for the elder populations in the USA[41,42]. Alum-formulated TLR4 agonist MPLA, referred to as Adjuvant System 04 (AS04), is already in clinical usage as part of the licensed human papillomavirus vaccine (Trade name: *Cervarix*) and hepatitis B vaccine (HBV) (Trade name: *Fendrix*)[42]. Very recently, CpG-1018, a TLR9a, has also achieved licensure in HBV (Trade name: *Heplisav-B*)[41,42]. Imiquimod, a TLR7a, as a cream (Trade name: *Aldara*) is licensed for treatment of external genital warts, keratosis and skin melanoma[43,44]. Alum-adsorbed TLR7/8a imidazoquinoline is currently employed in an inactivated COVID-19 vaccine (Trade name: *Covaxin*), which had been approved for emergency use in India[43,45]. Despite the impressive success of such adjuvanted vaccines, there are vulnerable populations for which current vaccines, including those employing adjuvants, do not achieve sufficient immunogenicity for protection[3,41]. Indeed, inadequate responses to vaccines disproportionally affect newborns and elders[4,46]. To understand and overcome such significant limitations, we applied mass cytometry (CyTOF) in combination with computational tools to reveal distinct, age-specific innate immune cell compartment in newborns, adults, and elders after PRRa stimulation.

In this study, we used a model of in vitro stimulation of age-specific human BMCs and CyTOF technology to map the early innate immune changes induced by various PRR agonists. To achieve a single-cell systems-level perspective of immune ontogeny that integrates age-specific cellular and molecular interactions in responses to PRRa, we leveraged single-cell mass cytometry along with DR analyses by the Cytobank platform to study 31 innate and adaptive immune cell populations in BMCs from healthy newborns, adults and elder individuals. Age-specific BMCs were stimulated in vitro with Alum, MPLA (TLR4 agonist (TLR4a)), R848 (TLR7/8a) or CpG (TLR9a) and stained with a 37-marker panel. By DR analysis, we found R848 outperformed all the selected PRRa and induced robust innate IFNγ responses in CD45$^+$ CD66a$^-$ mononuclear cells (MNCs) from all age groups, and the cytokine production relied predominantly on NK cells. Apart from NK cells, γδ T cells in adults contributed considerably to early IFNγ responses compared to newborns and elders. Additionally, clustering analysis confirmed the TLR7/8a-mediated effector γδ T cells abundance in adults. CD4$^+$ T cells also showed better innate IFNγ responses in elder only, upon R848 stimulation. Therefore, innate immune responses vary among age groups and may shape adaptive immunity in an age-specific manner.

## Results

### Age-specific mononuclear cells (MNCs) composition in human blood leukocytes

The relative abundance of major immune cell subsets in the blood is influenced by various factors such as age, sex, and previous infections in life[47]. To compare the composition of innate and adaptive immune cell subsets in immunologically distinct and vulnerable human populations, we mapped age-specific MNCs by running a 37-parameter CyTOF panel (Supplementary Table 1) and targeting 31 distinct immune cell populations (Supplementary Table 2). MNCs were isolated from cord blood (newborns) or peripheral blood (adults and elders), respectively. Healthy-term neonates (range: 38-40 weeks of gestational age), adults (range: 22-63 years), and elders (range: 65-85 years) were enrolled for this study (Supplementary Table 3 and Supplementary Data 2). In addition, the sample and data acquisition were completed before the COVID-19 pandemic was declared.

After the CyTOF run, Flow Cytometry Standard (FCS) files were exported to the cloud-based Cytobank platform[36–38]. The quality check step of exported cytometry files was done by the PeacoQC algorithm (Supplementary Fig. 1). PeacoQC removes signal shifts from issues like clogs, transient intensity changes, and acquisition speed changes[40]. We implemented DR analysis of quality-assured pre-gated MNCs, defined as live nucleated CD45$^+$CD66a$^-$ subset for visualizing datasets as well as the distribution of major immune cell populations in MNCs. A graphical representation of the manual gating strategy of MNCs in the Cytobank platform was shown in Supplementary Fig. 2. We took advantage of an optimized version of t-distributed Stochastic Neighbor Embedding (t-SNE) that uses Compute Unified Device Architecture (CUDA) enabled Graphics Processing Units (GPUs) to parallelize and speed up the computation[48]. By tSNE-CUDA representation and annotated clustering, we identified and measured the frequency of principal cell subsets participating in the innate [γδ T cells, natural killer T (NKT) cells, monocytes, myeloid dendritic cells (mDCs), plasmacytoid DCs (pDCs) and NK cells] and adaptive arm (CD4$^+$ T cells, CD8$^+$ T cells, and B cells) of the human immune system (Fig. 1, Supplementary Table 2 & Supplementary Fig. 3). Although tSNE-CUDA islands were similar in shape (Fig. 1A, Supplementary Fig. 3), their spreading was diverged and the relative abundance of γδ T cells and B cells differed to varying degrees between the groups (Fig. 1B). Cell frequencies demonstrated considerable intra- and inter-individual variability that were not specific for any cell population, an observation that is consistent with previous reports[47].

### PRRa-mediated innate activation of monocytes and DCs

Monocytes and DCs detect and eliminate pathogens in the early stages of infection. They also play a crucial role in the development of a protective immune response after vaccination. Activated DCs and monocytes are characterized by the induction of co-stimulatory surface molecules CD40

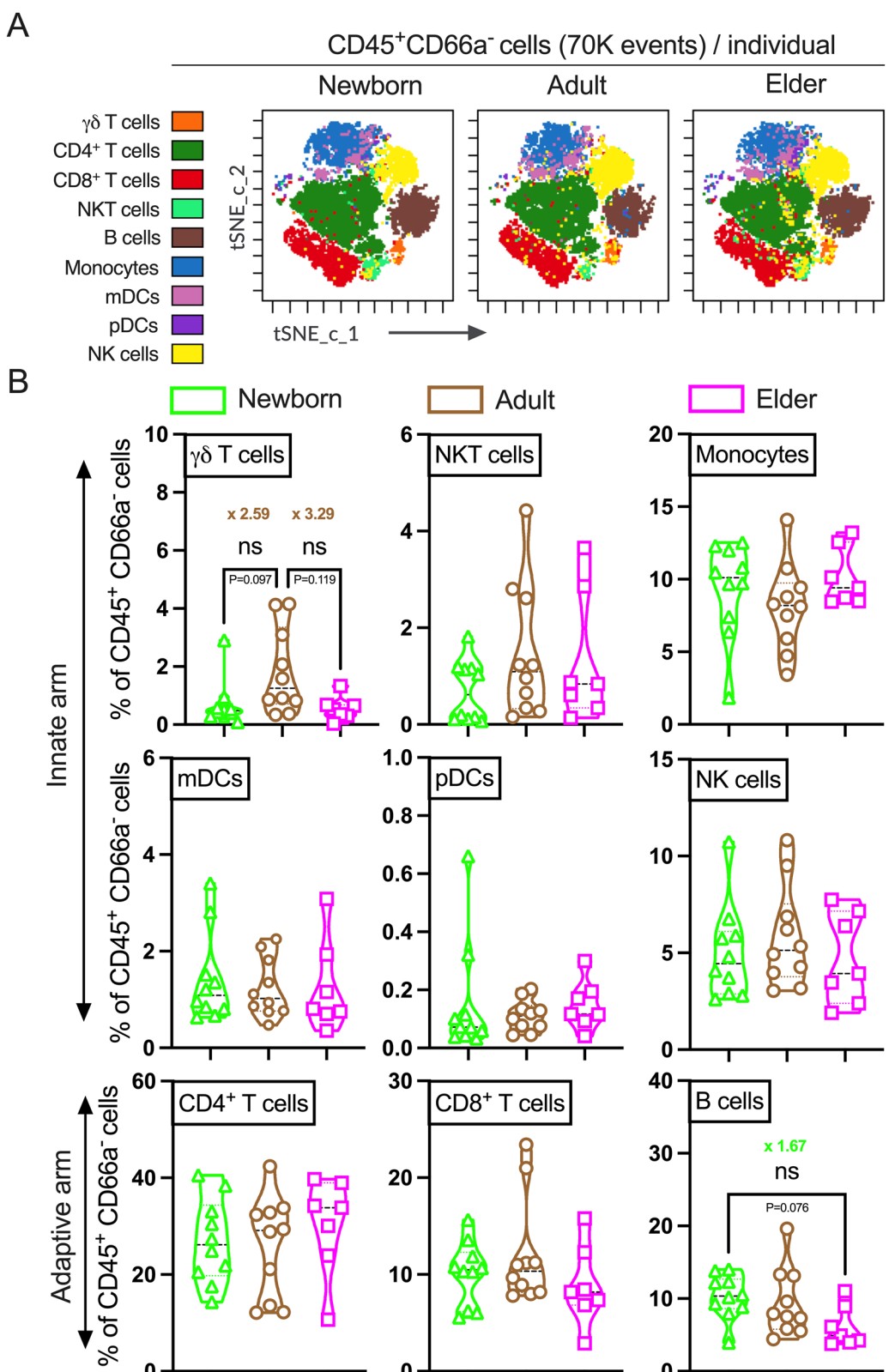

**Fig. 1 | Profiling of the MNCs (CD45$^+$ CD66a$^-$ cells) from newborns, adults and elders in unstimulated human blood leukocytes shows limited baseline differences. A** The tSNE-CUDA plots of representative newborn, adult and elder individuals showing the phenotypic distribution of indicated cell populations in MNCs. Indicated cell subsets were manually gated based on the phenotypic definition documented in Supplementary Table 2. For DR algorithm run, equal sampling (70K events of MNCs/donor) option was chosen in the Cytobank platform. tSNE-CUDA overlays identified manually gated cellular phenotypes and displayed by the indicated color profiles in the cellular landscape. **B** Violin plots showing the changes in the proportion of major immune cell subsets among CD45$^+$ CD66a$^-$ MNCs (exported 70K events of MNCs/donor) across the three cohorts from tSNE-CUDA overlays. Mean fold differences in γδ T cell and B cell compartments are shown. Each dot represents a single participant. Statistical comparison was performed using either one-way ANOVA or nonparametric Kruskal-Wallis test corrected for multiple comparisons; ns denoted non-significant. (n = 7–10 per group).

and CD86. Activation of CD40 on DCs helps in CD4$^+$ T cell priming via its ligand CD40L or CD154[49]. Expression of co-stimulatory molecule CD86 on antigen-presenting cells (APCs) is upregulated upon antigen encounter which facilitated the priming of T cells via CD28-mediated signaling[49]. After mapping the monocytes and DCs on newborn, adult, and elder cohorts, we focused on their activation stages upon PRRa adjuvants stimulation (Figs. 2, 3, Supplementary Figs. 4 & 5). Newborn and adult CD11c$^+$ monocytes showed superiority in CD86 induction upon CpG stimulation as compared to R848 and alum (Fig. 2D & Supplementary Fig. 4). The same trend was also portrayed on the CD11c$^+$ CD14$^-$ mDCs in adults and elders but not in newborn mDCs (Fig. 2A & B) or in CD11c$^-$ CD123$^+$ pDCs from adult and elder cohorts only when compared with alum stimulated groups (Fig. 2C). When focused on CD40 upregulation on DC lineages, we found that TLR7/8a R848 outperformed all the selected PRRa adjuvants (Fig. 3A–D & Supplementary Fig. 5). R848 showed its dominance in CD40 upregulation on DCs and age-specific divergence supporting previous studies by our group[14].

## Global effects of PRRa on innate IFNγ responses

Adjuvanted vaccines are generally recognized by the innate immune system and trigger adaptive immunity[42,50]. IFNγ orchestrates a key role in the activation of protective immunity against various infections[3,51]. IFNγ is also capable of triggering antitumor immune responses[52,53]. However, the role of adjuvant-induced IFNγ in shaping adaptive immunity upon vaccination has been overlooked. As such, to investigate the effect of PRRa on innate immune cell populations and effector cytokines like IFNγ secretion, BMCs from neonates, adults and elders were stimulated with adjuvants for 18 h. To visually inspect differences among groups of the samples without having to combine multiple files into a new FCS file, we took advantage of virtual concatenation (Figs. 4A-E) by the Cytobank platform. Figure 4 (along with Supplementary Figs. 6-8) showed that each PRRa induced a distinct, age-specific innate IFNγ response. tSNE-CUDA embedding identified a distinct island of IFNγ secreting innate immune cell lineage in the MNC compartment after R848 stimulation (Fig. 4E & Supplementary Figs. 6-8). Accumulation of IFNγ$^+$ MNCs upon R848 stimulation was significantly elevated compared to control, alum and MPLA-stimulated groups in all cohorts (Fig. 5A-F). R848-mediated IFNγ induction was only significant over CpG in the adult cohort (Fig. 5B). We noted that IFNγ induction after R848 stimulation was higher than CpG in newborns (3-fold, Fig. 5A) and in elders (2.7-fold, Fig. 5C), despite an absence of statistical significance. These data suggest a unique and robust ability of R848 to induce upregulation of innate IFNγ in MNCs from all age groups.

## PRRa-mediated early cytokine responses in MNCs

After mapping distinct TLR7/8a-mediated innate age-specific IFNγ profile in MNCs, we focused on the secretion of innate type 1 IFN and pro/anti-inflammatory cytokine signatures after PRRa stimulation. Type 1 IFN (IFNα) plays a vital role in innate immunity against bacteria and other pathogens[54]. Lack of insufficient type 1 IFN during early infection might lead to the onset of critical COVID-19 pneumonia[55]. Th1 (TNF) and Th17 (IL-17) related cytokines robustly augment innate and adaptive immune response, thereby enhancing the efficacy of vaccines[3]. IL-23p19 has a Th1 induction activity[56]. Anti-inflammatory cytokine IL-10 plays an essential role in maintaining mucosal homeostasis[57,58]. Here, we measured these essential early cytokines production in response to vaccine adjuvants by CyTOF.

R848 significantly induced innate TNF responses over MPLA in both newborns and adults (Supplementary Fig. 9B). Furthermore, such induction was visible over alum-treated newborn MNCs but absent in adult and elder groups. R848 also triggered significant IL-23p19 responses over alum in newborns only (Supplementary Fig. 9C) but IL-10 responses over alum in both newborns and adults (Supplementary Fig. 9E). CpG-mediated early TNF responses were captured in newborns (significant when compared with alum) only (Supplementary Fig. 9B). However, our study was not able to detect any PRRa-mediated induction of IFNα and IL-17 in different age groups (Supplementary Fig. 9A & D). Remarkably, we captured TLR7/8-

mediated enhancement of IL-10 (3.56-fold increase, compared to control, Fig. 5F & Supplementary Fig. 9E) and IL-23p19 (5.66-fold increase, compared to control, Fig. 5F & Supplementary Fig. 9C) in elder's MNC compartment. Taken together, R848 induced a distinct, age-specific cytokine profile in MNCs (Fig. 5 & Supplementary Fig. 9). These observations highlight the potential of TLR7/8a as a candidate adjuvant to enhance immunogenicity of candidate vaccines for vulnerable populations.

## TLR7/8a-mediated abundance of NK cells in innate IFNγ$^+$ MNC compartment

TLR7/8a may enhance early life immunogenicity as an adjuvant and overcomes neonatal hyperresponsiveness to acellular pertussis vaccine[11] as well as pneumococcal conjugate vaccine[15] by inducing robust Th1 cytokines. After seeing the robust induction of Th1-related cytokine IFNγ by TLR7/8a R848 (Fig. 5), we tried to capture the predominant source of IFNγ production mediated by major innate cell lineages among MNCs. IFNγ, a type II interferon, is critical for innate and adaptive immunity against viral, bacterial and protozoan infections[54,59,60]. Figure 6A shows the predominant island of IFNγ secreting MNCs in tSNE-CUDA plots. By tSNE-CUDA embedding, we overlaid all the manually gated major innate and adaptive immune cell lineages (according to Supplementary Table 2) among R848 stimulated MNCs and visually found the IFNγ secreting predominant island belongs to the NK cell island colored by yellow (Fig. 6B). tSNE-CUDA embedding identified a statistically significant abundance of NK cells among IFNγ$^+$ MNCs (Fig. 6D, Supplementary Fig. 10 and Supplementary Fig. 11A), which were universal across all cohorts (Supplementary Fig. 11B). To get more insight of the innate IFNγ secreting NK cells upon R848 stimulation, we ran another tSNE-CUDA by targeting 2500 IFNγ$^+$ MNCs per participant (Fig. 6C). tSNE-CUDA embedding after down-sampling also confirmed the similar abundance of NK cells compared with the prior tSNE-CUDA run (Fig. 6C and Supplementary Fig. 11C). We did not find any age-specific compositional differences in B cells, mDCs, pCDs, monocytes, NKT cells and CD8$^+$ T cells pertaining to IFNγ producing MNCs after R848 stimulation (Supplementary Fig. 11D).

## TLR7/8a-mediated unique abundance of IFNγ producing γδ T cells in adult MNCs

Having demonstrated the age-universal effect of TLR7/8-mediated IFNγ production in NK cells, B cell, mDC, pDC, monocyte, NKT cell, and CD8$^+$ T cell compartment, we focused on CD4$^+$ T cell and γδ T cell subsets. CD4$^+$ T cells and γδ T cells were manually gated as MNC$^+$ CD3$^+$ CD19$^-$ TCRgd$^-$ CD11c$^-$ CD14$^-$ CD4$^+$ CD8$^-$ and MNC$^+$ CD3$^+$ CD19$^-$ TCRgd$^{dim, +}$ CD14$^-$ CD4$^-$ CD8$^-$ respectively (Supplementary Table 2). Both subsets were subjected to dedicated tSNE-CUDA embedding upon R848 stimulation (Fig. 6, Supplementary Fig. 10 and Supplementary Fig. 11A). We identified an age-specific abundance of CD4$^+$ T cells (Fig. 7A) and γδ T cells (Fig. 7B) in the IFNγ$^+$ MNC compartment upon R848 stimulation. The abundance of CD4$^+$ T cells was found to be higher in elders (Fig. 7A), whereas γδ T cells were higher in adults (Fig. 7B) upon R848 stimulation. To investigate differences within CD4$^+$ T cells and γδ T cell populations across age groups, Spanning Tree Progression Analysis of Density Normalized Events (SPADE)[61] was implemented using the Cytobank platform. SPADE distilled high-dimensional data down to interconnected nodes to further the analysis of cellular heterogeneity, identification of cell subsets and comparison of functional markers in response to stimuli. The size and the color of the circular nodes correlates with cell number (Fig. 7A and B, right panel; Supplementary Fig. 12G-I). Although most of these nodes failed to be distinct in CD4$^+$ T cell compartment (Fig. 7A; right panel & 7C), nodes 16, 22, and 30 showed significant abundance in adult participant's γδ T cell compartment (Fig. 7B; right panel & 7D). Consistent with a previous report[62], γδ T cells exist as heterogeneous populations with distinct properties. We further immunophenotyped nodes 16, 22, and 30 according to the expression of memory phenotype-defining markers CD45RA, CD27 and CCR7[63,64] along with γδ T cells marker TCRgd (Supplementary Fig. 13). As speculated, TCRgd expression is high throughout the nodes

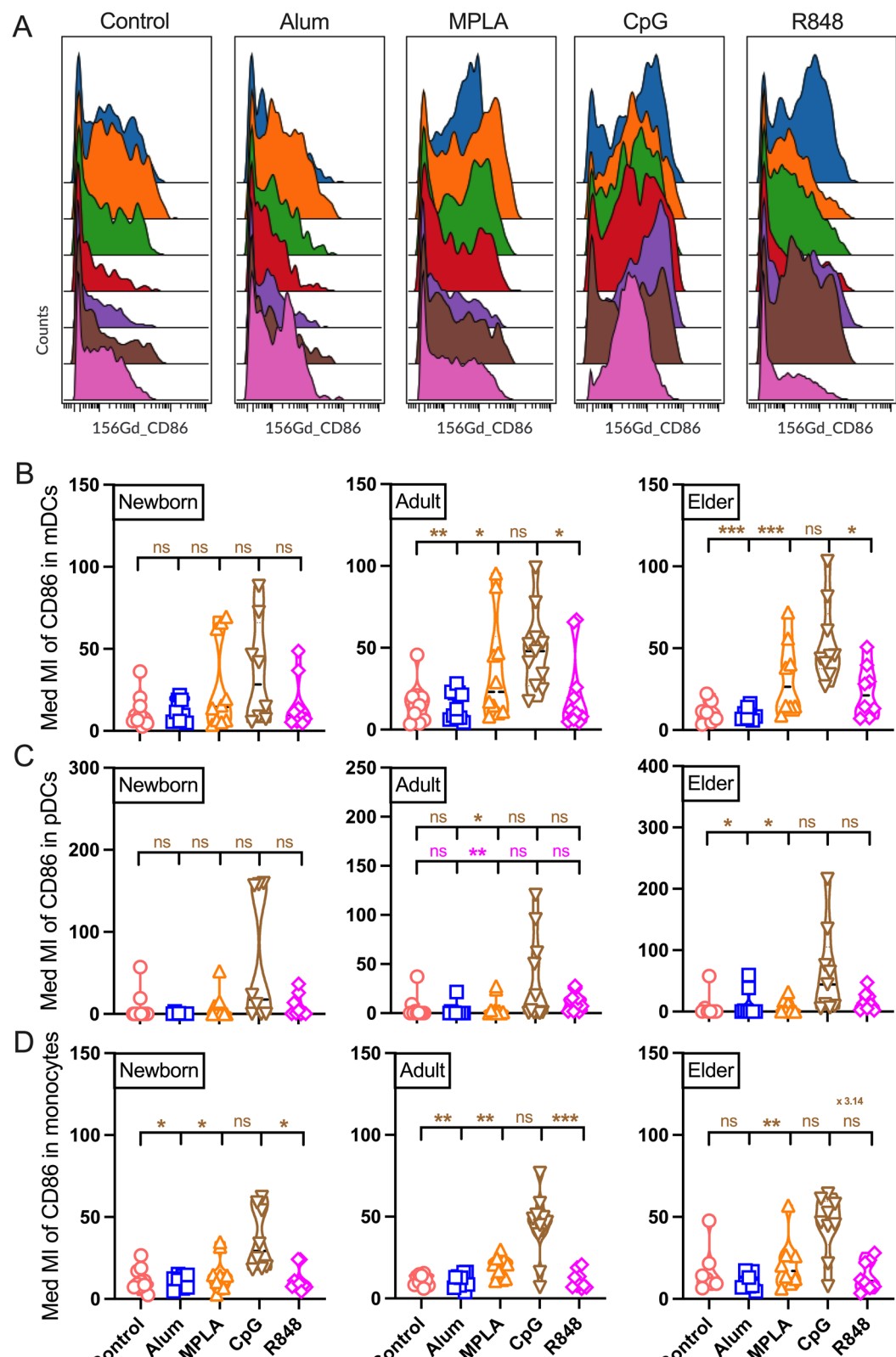

**Fig. 2 | Activation profile of co-stimulatory molecule CD86 on mDCs, pDCs and monocytes after PRRa stimulation. A** Representative median metal intensity (Med MI) of CD86 expression on mDCs in the elder cohort. Color in histogram indicates each participant's ($n = 7$) CD86 activation profile upon PRRa stimulation. Non-stimulated BMCs (control) or stimulated with alum (10 µg/ml), MPLA (100 ng/ml), CpG (5 µM) and R848 (5 µM) for 18 h from newborn, adult and elder cohorts. Med MI of (**B**) mDCs, (**C**) pDCs and (**D**) monocytes are shown for each stimulation. Mean fold difference between CpG and R848 in elder monocyte compartment is also shown. Statistical comparison was performed using either one-way ANOVA or nonparametric Kruskal-Wallis test corrected for multiple comparisons; *$p < 0.05$, **$p < 0.01$, ***$p < 0.001$, ns denoted non-significant. Each dot represents a single participant ($n = 7$-11 per group).

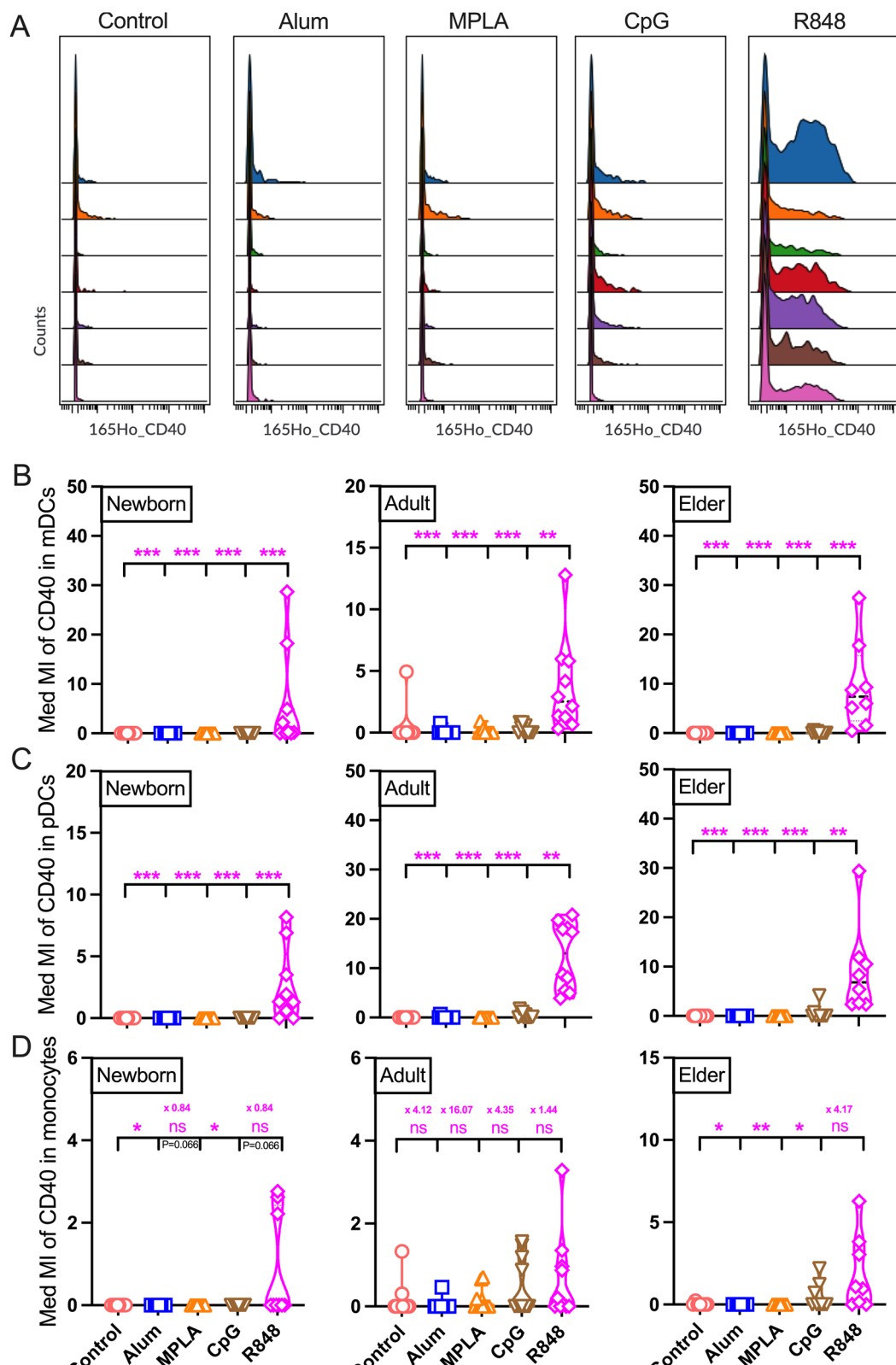

**Fig. 3 | TLR7/8a R848 drives CD40 upregulation on DCs and monocytes.**
**A** Representative median metal intensity (Med MI) of CD40 expression on mDCs in the elder cohort. Color in the histogram indicates each participant's ($n = 7$) CD40 activation profile upon PRRa stimulation. Non-stimulated BMCs (control) or stimulated with alum (10 μg/ml), MPLA (100 ng/ml), CpG (5 μM) and R848 (5 μM) for 18 h from newborn, adult and elder cohorts. Med MI of (**B**) mDCs, (**C**) pDCs, and (**D**) monocytes are shown for each stimulation. Mean fold differences between R848 and rest PRRa adjuvants in monocyte compartment are also shown. Statistical comparison was performed using nonparametric Kruskal-Wallis test corrected for multiple comparisons; *$p < 0.05$, **$p < 0.01$, ***$p < 0.001$, ns denoted non-significant. Each dot represents a single participant ($n = 7$–11 per group).

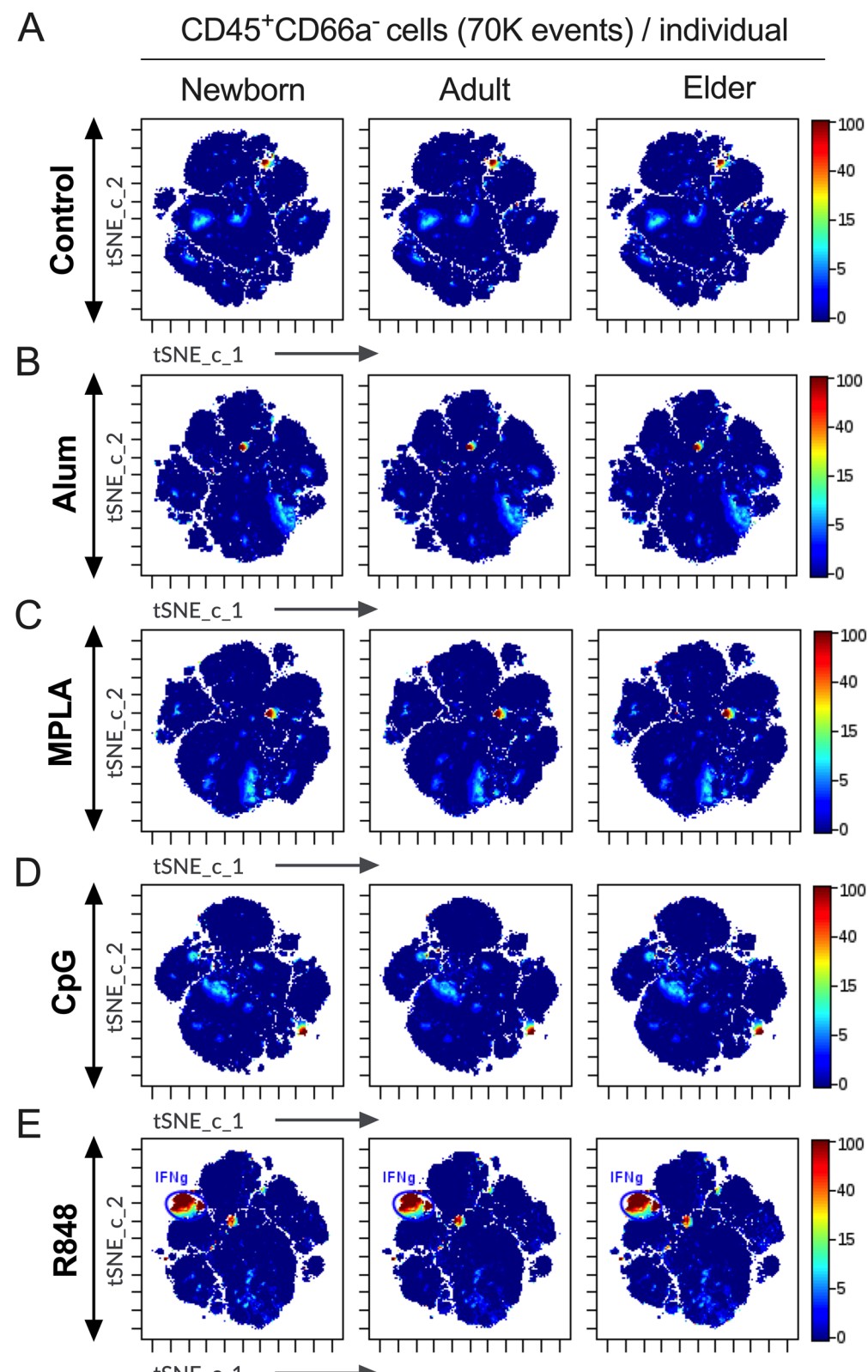

**Fig. 4 | Innate IFNγ response in MNCs is distinct upon TLR7/8a treatment during PRRa adjuvant screening.** IFNγ expression (in Z-axis channel) overlaid on tSNE-CUDA embedding. Color indicates median metal intensity of IFNγ expression ranging from low (blue) to high (red). 70K events of MNCs from each donor used in concatenated visualization with tSNE-CUDA. **A** Non-stimulated BMCs or stimulated with (**B**) alum (10 μg/ml), (**C**) MPLA (100 ng/ml), (**D**) CpG (5 μM), and (**E**) R848 (5 μM) for 18 h from the newborn, adult and elder cohorts (*n* = 7–11 per group). For R848 stimulation, a manual gate in tSNE-CUDA plot indicates (for visualization, not for quantification) the predominant island expressing IFNγ in MNCs.

**Fig. 5 | R848 has greater IFNγ inducing efficacy than other PRRa in human MNCs.** Non-stimulated BMCs were treated as control groups. BMCs were stimulated with alum (10 μg/ml), MPLA (100 ng/ml), CpG (5 μM) and R848 (5 μM) for 18 h from (**A**) newborn, (**B**) adult and (**C**) elder cohorts. **A–C** Proportion (frequencies) of IFNγ⁺ cells in MNCs are shown for each stimulation. Mean fold differences between R848 and MPLA in addition to CpG are also shown. **D–F** Radar plot demonstrated mean fold changes of cytokine levels (frequencies of cytokine⁺ cells in MNCs) relative to vehicle control. Statistical comparison was performed using non-parametric Kruskal-Wallis test corrected for multiple comparisons; *$p < 0.05$, **$p < 0.01$, ***$p < 0.001$, ns denoted non-significant. Each dot represents a single participant ($n = 7$–$11$ per group).

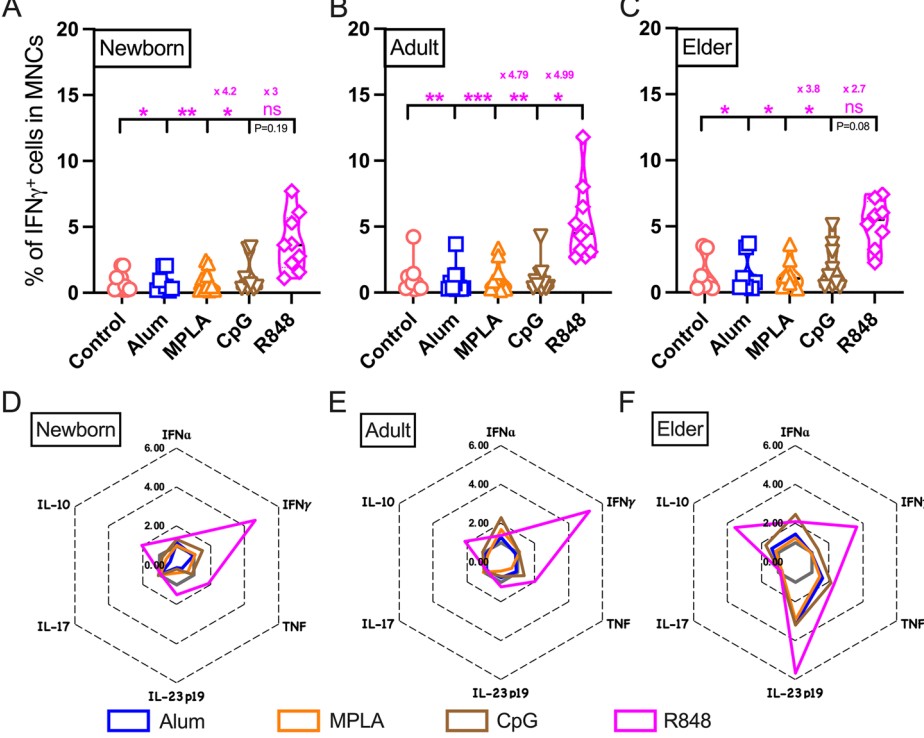

(Supplementary Fig. 13B). Low expression of CD45RA (Supplementary Fig. 13C) and variable (low to high) expression of CD27 (Supplementary Fig. 13D) defined a mixed central memory (CD45RA^low CD27^high), and effector memory (CD45RA^low CD27^low) phenotypes among nodes 16, 22 and 30. Conclusively, CCR7 upregulation in node 30 defined a central memory (CD45RA^low CCR7^high) phenotype (Supplementary Fig. 13E), whereas nodes 16 and 22 consisted of effector memory (CD45RA^low CCR7^low) γδ T cells. The abundance of cytotoxic CD56⁺ γδ T cells were observed in node 16 when compared with node 30 (Supplementary Fig. 13F). Cells in node 16 which are IFNγ producing CD56⁺ effector γδ T cells might possess cytotoxic activity classically associated with anti-tumor responses[65,66]. Taken together, these data indicate that the functional plasticity of the γδ T cells is influenced by age and stimulus type.

## Discussion

The goal of this study was to evaluate the activity of PRRa as vaccine adjuvants towards human BMCs derived from newborn, adult and elder individuals with respect to cellular composition and cellular sources of innate cytokine production, with particular focus on production of IFNs, which are important to both host defense and vaccine responses. We took advantage of age-specific human in vitro modeling merged with machine learning algorithmic tools like tSNE-CUDA and SPADE, which are developed for DR and clustering to handle the multiparameter data complexity. The high degree of granularity provided by computational tools identified immune cell lineage differences among age groups. The immune atlas thus developed offers new avenues to dissect age-specific functional immunity – a key tool for the development of age-tailored vaccine adjuvants[1,4].

APCs and PRRa adjuvants play an important role in the immune response to pathogens and vaccine formulations. TLR7/8a-mediated activation of CD40 on pDCs in human newborn and adult blood has been reported by our group in the past[14]. Here we showed that CD40 expression on DCs and monocytes was affected by TLR7/8a stimulation whereas CD86 expression was enhanced by TLR9a. Successful antigen presentation requires the activation of the monocytes and DCs via co-stimulatory molecules CD40 and CD86. Therefore, we hypothesized that TLR7/8a and

TLR9a could be beneficial to unleash the maximum immunogenicity of the targeted vaccines across the human lifespan. Furthermore, experience with several adjuvanted vaccines employing TLR agonists, including *Covaxin*, an Alhydroxiquim-II (alum formulated TLR7/8a) adjuvanted vaccine, as well as *Heplisav-B* and *Corbevax* both CpG-1018 (TLR9a) adjuvanted vaccines, has been encouraging. These vaccines have been administered to >150 million individuals worldwide, many of whom were older adults, with safety surveillance data suggesting general safety and any phenomena that could potentially be attributed to exacerbation of inflammaging are rare[45,67–69].

By mapping innate and adaptive immune cell lineages among PRRa-stimulated BMCs, we found that early IFNγ responses could aid in better prediction of TLR7/8a adjuvanticity. TLR7/8a adjuvant R848 induced robust IFNγ responses in MNCs from all age groups. It should be noted that neither alum nor MPLA or CpG induced substantial concentrations of IFNγ in any age groups. Further, R848 induced cytokines such as IL-23p19 and TNF, along with anti-inflammatory IL-10. Although CpG induced moderate production of TNF in newborns (when compared with alum), this effect was reduced in adults and elders. Thus, our precision approaches of age and adjuvant-specific activation of MNCs and dissecting innate cytokine responses by immunophenotyping may help to identify age-specific adjuvants to design next-generation vaccine formulations.

By dissecting the innate IFNγ compartment after PRRa stimulation, we demonstrated that NK cells contributed relatively more (when compared with other immune cell lineages) to TLR7/8a-mediated total IFNγ production in newborn, adult, and elder groups. Several studies showed that R848-mediated IFNγ induction by NK cells follows the IL-2/IL-12 axis[70–72]. Such boosters in effector NK cells might unleash a new strategy for cancer immunotherapy and infectious disease therapy[73]. NK cells were described as the largest population of splenic IFNγ⁺ cells during the course of influenza A virus infection in mice[74]. Remarkably, another recent study demonstrated the pleiotropic roles of cytokine-producing NK cells during influenza virus infection in humans[75]. Further studies are needed to understand the complicated mechanisms of PRRa-mediated NK cell activation.

Furthermore, γδ T cells contributed relatively more to total IFNγ production in adults, whereas CD4⁺ T cells contributed more to elders. Both

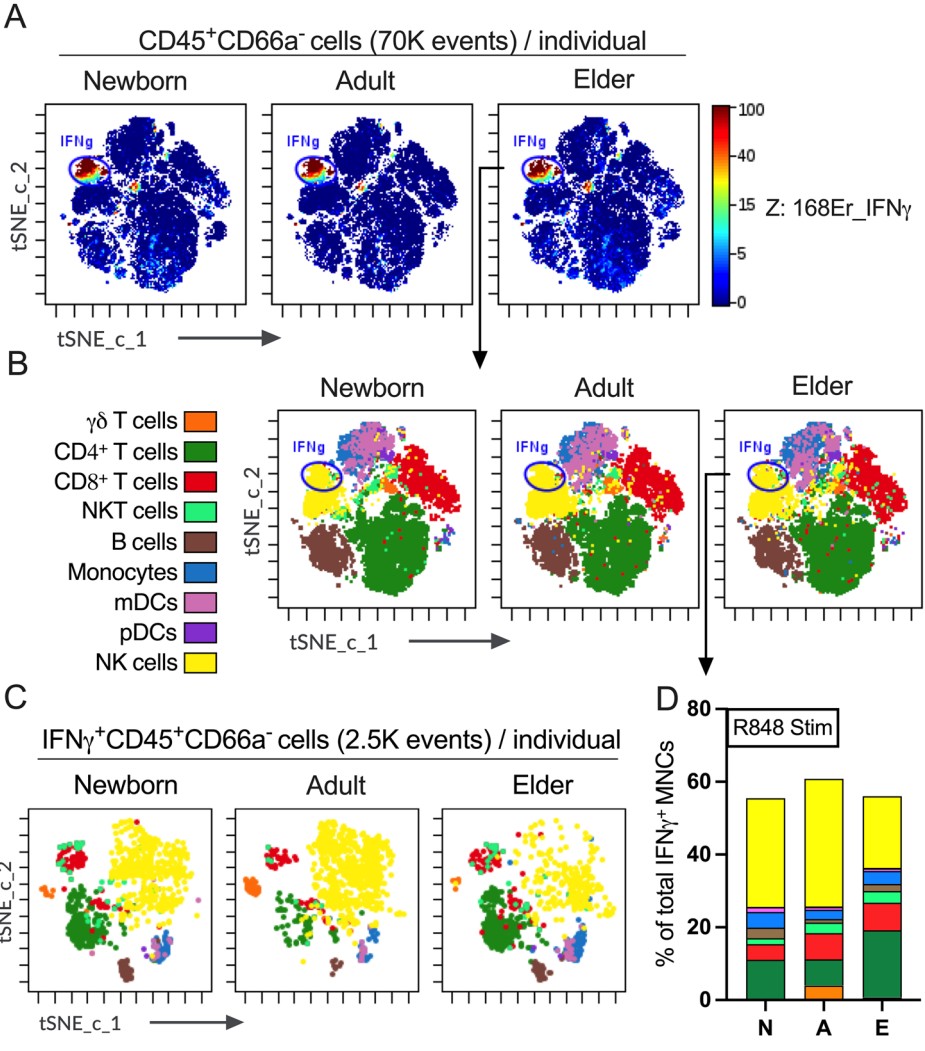

**Fig. 6 | Mapping the cellular abundance of IFNγ producing MNCs by tSNE-CUDA reveals a unique role of NK cells. A** Representative plot of IFNγ expression (in Z-axis channel) of MNCs after R848 stimulation was overlaid on tSNE-CUDA embedding. Color indicates IFNγ expression ranging from low (blue) to high (red). A manual gate in tSNE-CUDA plot indicates (for visualization, not for quantification) the predominant island expressing IFNγ in MNCs, which fall into (**B**) tSNE-CUDA island of NK cells. **C** tSNE-CUDA overlays comparing the topography of tSNE plots from (**B**) after downsampling of 2500 IFNγ⁺ MNCs per donor from newborn, adult and elder cohorts. **D** Comparison between the frequency (as a percentage of the IFNγ⁺ MNCs) from the tSNE-CUDA overlaid embedding (from **B**) of the major immune cell lineages after R848 stimulation to find the key source of innate IFNγ production (*n* = 8–10 per group). N stands for newborn; A, adult and E, elder.

tSNE-CUDA and cluster analysis by SPADE showed a distinct profile with functional plasticity of γδ T cells in adult participants after R848 stimulation. Even without stim, the accumulation of γδ T cells was found to be higher in adults compared with newborns (~2.5 fold) and elders (~3 fold, Fig. 1B). Age-associated alterations in γδ T cell compartment are well studied[76–78]. γδ T cells showed an association during viral infections like Cytomegalovirus[79,80], influenza[81] and COVID-19[82–84]. BCG vaccination triggers robust γδ T cell responses in adults[18]. Therefore, an age-tailored adjuvantation system for newborn and elder might be beneficial for targeting γδ T cells rather than classical innate immune cells.

Importantly, elderly CD4⁺ T cells demonstrated even greater responsiveness upon R848 stimulation than those of adults, suggesting that some T cell subsets may assist with early IFNγ production in this age group. SPADE analysis did not find any heterogeneity among twelve functional nodes in CD4⁺ T cell compartments. Early activation of aged CD4⁺ T cells might lead to contraction though the range of innate stimulation for which this pattern holds remains to be defined. We did not find any innate like TLR7/8 activation in B cell compartment. Consistent with prior studies[85,86], we captured ontogeny-specific reduction of B cells in the elder group. Therefore, innate immune responses vary among age groups and have the potential to shape adaptive immunity in age-specific manners.

Our study demonstrates the feasibility and advantages of using multiparametric analyses in characterizing age-specific differences in MNCs. The effect of age on composition, phenotype, and function of MNCs, was holistically captured by a mix of supervised and unsupervised computational tools developed for multiparametric analyses of single cells. Finding immune responses mediated by innate cell lineages that correlate with protection are key to accelerating the development of novel age-tailored vaccines. This study also establishes CyTOF as a powerful tool to characterize immune ontogeny as well as adjuvant/adjuvanted vaccine-induced immune responses.

Our study features several strengths, including (a) the immuno-phenotypic characterization of the human MNC compartment by age; (b) use of in vitro screening of multiple PRRa for functional activity toward MNCs from human neonates and elders as compared to middle-aged adults; and (c) application of singe-cell mass cytometry to study cell interactions and functional output in freshly isolated MNCs from neonates, adults and elders directly compared side by side. Overall, our study may inform the rational design of adjuvanted vaccine formulations for age-tailored immunization.

Along with its multiple strengths, our study also has some limitations, including (a) our study characterized MNCs at a single time point with single concentration for selected adjuvants, providing a snapshot of cellular composition and function; (b) neonatal modeling employed cord blood which, given the rapid changes of composition, phenotype, and function of immune cells occurring in the first week of life[87,88], may not fully reflect the immune status of a newborn after birth, though our prior in vitro human cord blood studies accurately predicted activity of PRR-targeting adjuvants in vivo[15]; (c) our study was not designed to investigate other relevant host factors such as microbiota, genetics or immune imprinting and lastly (d)

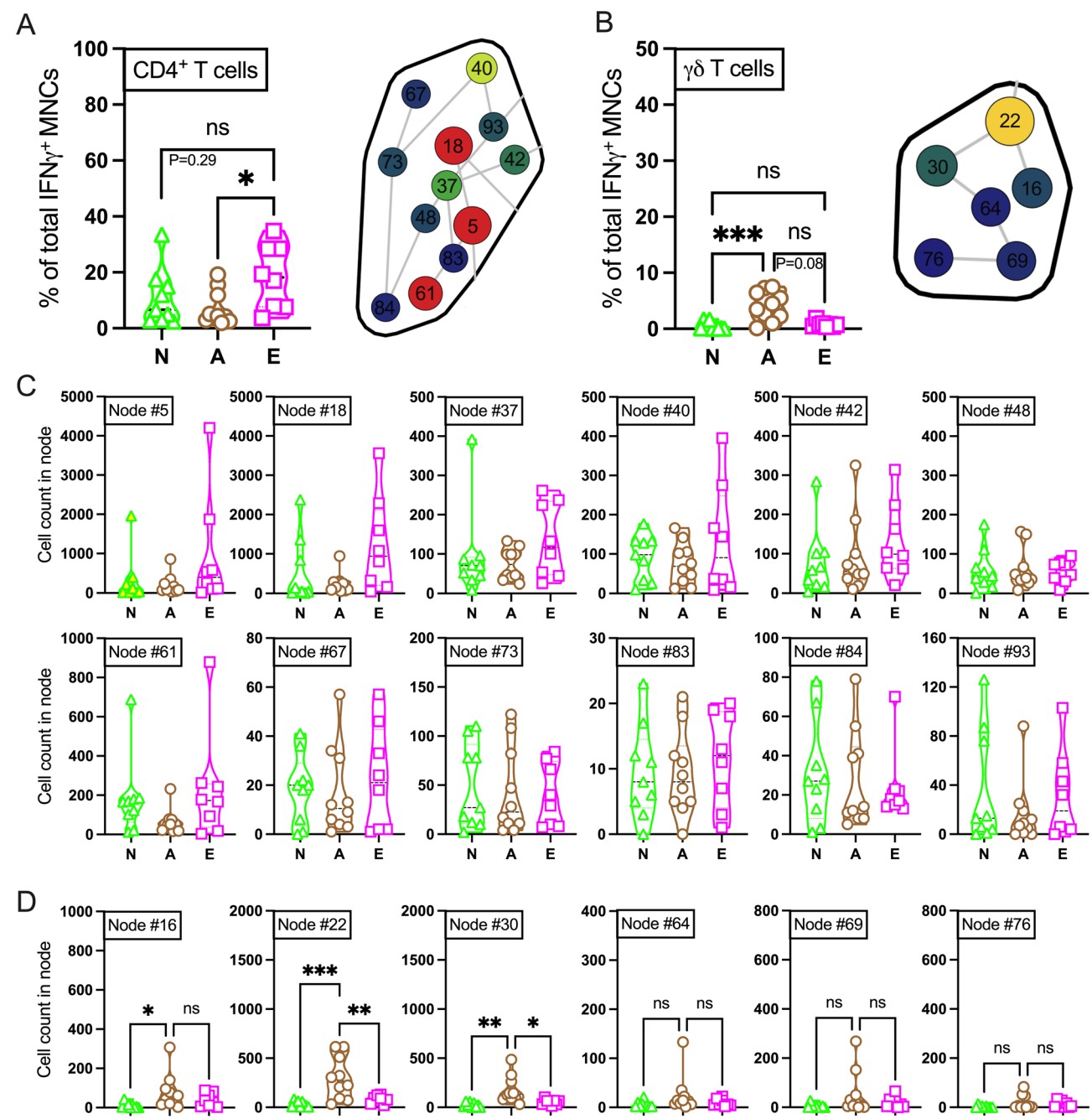

**Fig. 7 | SPADE analysis of significantly abundant immune cell lineages in IFNγ producing MNC compartment after TLR7/8a (R848) stimulation highlights a secondary role of T cell subsets.** Proportion (frequencies) of (**A**) CD4+ T cells and (**B**) γδ T cells in IFNγ+ MNC compartment after R848 stimulation (N stands for newborn; **A**, adult and **E**, elder) from the tSNE-CUDA overlaid embedding (Fig. 6B & D). R848 stimulated IFNγ+ MNCs from all cohorts were used to generate SPADE trees by the Cytobank platform. Representative SPADE bubble of manually annotated CD4+ T cells (**A**, right panel) and γδ T cells (**B**, right panel) from elder (Supplementary Fig. 12I) and adult (Supplementary Fig. 12H) participants, respectively, shown here. Cell abundance is represented by size of the nodes. Each node displays a gradient from low (blue) to high (red) depending on the respective number of clustered cells (from Supplementary Fig. 12H & I). Indicated node IDs were generated anonymously by the Cytobank platform. **C** All the nodes from SPADE bubble of CD4+ T cells and (**D**) γδ T cells were analyzed to detect dependence of cellular abundance with age. Each dot represents a single participant (n = 8–10 per group). Statistical comparison was performed using either one-way ANOVA or nonparametric Kruskal-Wallis test corrected for multiple comparisons; *p < 0.05, **p < 0.01, ***p < 0.001, ns denoted non-significant.

even though our team has extensively demonstrated the in vivo potential of TLR7/8a adjuvants[89], we focused on hypothesis generation via in vitro modelling for this study. Future studies, including those employing high-dimensional single-cell analyses, will further characterize phenotypical and functional changes, further informing the discovery and development of adjuvanted vaccines to protect vulnerable populations such as infants and elders.

## Methods
### Study participants
Thirty individuals were included in this study (Supplementary Table 3 and Supplementary Data 2). Blood samples were collected in between June 2018 and September 2018. Peripheral blood was collected from healthy adult (n = 11) and elder volunteers (n = 9), while human newborn cord blood (n = 10) was collected immediately after Cesarean

section delivery of the placenta. Births to known HIV-positive mothers were excluded.

## Study approval

Nonidentifiable human cord blood samples were collected with the approval from the Ethics Committee of the Brigham & Women's Hospital, Boston, MA (Institutional Review Board (IRB) protocol number 2000-P-000117), and Beth Israel Deaconess Medical Center Boston, MA (IRB protocol number 2011P-000118). The requirement for informed consent was waived for the non-identifiable human cord blood samples. Peripheral blood from healthy elder donors was obtained from the outpatient clinics of Brigham & Women's Hospital (Boston, MA) after written informed consent with approval from the IRB (protocol no. 2013P002473). Blood from healthy, adult volunteers was collected after written informed consent with approval from the Ethics Committee of Boston Children's Hospital, Boston, MA (protocol number X07-05-0223). All ethical regulations relevant to human research participants were followed.

## Human blood sample processing and in vitro stimulation

Human blood was anticoagulated with 20 units/ml pyrogen-free sodium heparin (American Pharmaceutical Partners, Inc.; Schaumberg, IL). All blood products were kept at room temperature (RT) and processed within 4 h from collection. For human blood mononuclear cells (BMCs) stimulation, primary human BMCs were isolated from fresh blood via Ficoll gradient separation. BMCs were resuspended and maintained in RPMI 1640 culture media (Invitrogen, Grand Island, NY) containing 10% autologous plasma. Cells were plated at $4 \times 10^6$ cells/ml in 24-well plates and stimulated with aqueous formulations of selected PRRa (Supplementary Table 3) at the concentrations noted in the figure legends. The concentration of PRRa for in vitro stimulations were chosen empirically based on either the results of previous[11,15] studies or according to the manufacturer's (Invivo-Gen) instruction. After 10 h of incubation in a humidified incubator at 37 °C (in 5% $CO_2$), GolgiPlug (containing Brefeldin A) and GolgiStop (containing Monensin) were added according to the manufacturer's (BD Biosciences) instructions to all the wells to block the cytokines production and facilitate optimal intracellular mass cytometry analysis. Cultures were maintained in a humidified incubator at 37 °C (in 5% $CO_2$) for an additional 8 h.

## Mass cytometry

Information about the antibodies and reagents used for CyTOF staining are provided in Supplementary Tables 1 & 3. Pre-conjugated metal-tagged antibodies were purchased from Standard BioTools (formerly, Fluidigm). Custom conjugation of certain antibodies (specifically, IFNα, CD80, and TCRγδ) were accomplished at Harvard Medical Area CyTOF Antibody Resource and Core (Boston, MA) according to the Core's optimized protocol. After 18 h of stimulation, cells were harvested and washed twice with Maxpar® PBS. Up to $3 \times 10^6$ cells per stimulation were stained with Cell-ID Cisplatin-195Pt (viability stain) and blocked with Human TruStain FcX™ according to the manufacturer's instructions. After blocking, cells were incubated with a cocktail of metal-tagged surface markers for 30 minutes at RT. Cells were washed with Maxpar® Cell Staining Buffer and fixed using the Maxpar® Fix I Buffer according to the manufacturer's instructions. Cells were washed with Maxpar® Barcode Perm Buffer and barcoded using the Cell-ID 20-Plex Pd Barcoding Kit according to the manufacturer's instructions. After barcoding, samples from all the stimulating conditions were combined, washed twice with Maxpar® Perm-S Buffer, and subjected to intracellular staining (30 minutes at RT) using a cocktail of the seven metal-conjugated antibodies (Supplementary Table 1). After washing twice with Maxpar® Cell Staining Buffer, stained cells were incubated at 4 °C overnight with a DNA intercalator (Cell-ID Intercalator-Ir). The following morning, cells were washed with Maxpar® Cell Staining Buffer followed by Maxpar® Cell Acquisition Solution (CAS), filtered through a 35 μm cell strainer and resuspended in Maxpar® CAS containing 0.1× EQ Four Element Calibration Beads at a cell concentration of $1 \times 10^6$ cells/ml. Cells were run on the Helios (Standard BioTools) instrument at Longwood Medical

Area CyTOF Core at Dana-Farber Cancer Institute (DFCI) according to the standardized good clinical laboratory practice procedures[90,91].

## Mass cytometry data processing and quality assessment

After acquisition by the Helios instrument, samples were normalized and de-barcoded according to manufacturer's recommendation with CyTOF software (version 6.7.1014)[92]. Normalized and de-barcoded FCS files were uploaded to the Cytobank platform (version 10.3) (Beckman Coulter Inc., CA)[37]. Prior to immunophenotyping, anomalous events were removed by PeacoQC algorithm[40] as described in Supplementary Fig. 1. For PeacoQC run, the recommended setting in the Cytobank platform was followed (Remove margins: OFF; Anomaly detection method: Both; MAD parameter: 6; IT limit parameter: 0.6; Max bins: 500 and Consecutive bins: 5). After removal of anomaly, quality-controlled FCS files were further cleaned up by removing EQ calibration beads and gating live (195Pt⁻) nucleated (191Ir⁺ and 193Ir⁺) cell events (Supplementary Fig. 2). Live nucleated cells were manually gated further to map 31 distinct populations as described in Supplementary Table 2. Cell gating strategy was adopted from[25,27,93]. Five markers were excluded during quality assessment due to no signal or technical errors (during the experiment) from downstream analysis (Supplementary Data 2).

## DR analysis of mass cytometry data

tSNE-CUDA[48] and SPADE[61] analyses were performed using the Cytobank platform. For each tSNE-CUDA run 70K events of MNC population (CD45⁺CD66a⁻) were sampled equally. The FCS files are automatically excluded from the tSNE-CUDA analysis which does not possess 70K events of MNC population. For each tSNE-CUDA algorithm run, the parameters were iterations = 1500, perplexity = 50, learning rate = 166,500, and early exaggeration = 12. We included all the 32 channels except 89Y, 171Yb, 191/193 Ir, and 195 Pt (Supplementary Table 1) for DR analysis using tSNE-CUDA. For Fig. 6C, 2500 events of IFNγ⁺ MNC⁺ effector cell subset were selected for DR analysis, and the learning rate was 5146. From the tSNE-CUDA landscapes, all the desired cell populations were calculated and graphed using GraphPad Prism version 10.2 for macOS for immunophenotyping.

For clustering analysis, we ran the SPADE algorithm in R848 stimulated IFNγ⁺ MNC⁺ effector cell subset using the following parameters: target number of nodes = 100 and downsampled events target = 10%. For the clustering channel, 15 cell surface and intracellular markers were selected as follows: CD20, CD123, CD16, CD19, CD4, CD11c, CD14, IFNγ, CD45RA, CD3, CD38, HLA-DR, CD8, TCRγδ and CD56. The clustering channels were selected based on the cell populations to be clustered. CD4⁺ or CD8⁺ T cells, γδ T cells, NK cells, B cells, monocytes, and pDCs were bubbled manually by using an overlaid expression pattern of selected clustering channels (Supplementary Fig. 12). Cell abundances in the bubbled CD4⁺ T cells and γδ T cells were further dissected by cell counts within each node by the Cytobank platform. Next, we exported nodes 16, 22, and 30 from the SPADE bubble of R848 stimulated adult IFNγ⁺ γδ T cells using the Cytobank platform and portrayed the phenotypic differences based on memory signature (CD45RA, CD27 and CCR7) along with functional (CD56) profile.

## Statistics and reproducibility

Statistical significance and graphic output were generated using the GraphPad Prism version 10.2 for macOS (GraphPad Software, La Jolla, CA). Data are represented in violin plots as the median with interquartile range. Data were tested for normality using the Shapiro-Wilk test. Group comparisons were performed by one-way ANOVA followed by post hoc Tukey's test for multiple comparisons. Measurements that failed normality tests were analyzed with a Kruskal-Wallis rank-sum test followed by Dunn's multiple comparison within treatment groups. 2-tailed paired t-test was used for comparison between 2 groups for Supplementary Fig. 11C. Results were considered significant at p-values indicated in each figure legend. The sample size for in vitro stimulations were chosen empirically based on the

results of previous studies[11,15]. In addition, for each CyTOF experiment (a total of 10 independent assays with BMCs from 30 participants, Supplementary Data 2) matching representatives of newborn, adult and elder samples were prepared to reduce batch effects, and each experiment was performed as a single assay.

## Reporting summary

Further information on research design is available in the Nature Portfolio Reporting Summary linked to this article.

## Data availability

Data are available in the main article, figures, tables, supplementary materials and in supplementary Excel files, which are available online with this article. Values for all data points in graphs and mean fold changes between groups of interest are reported either in Supplementary Figs. or in the separate Excel file named "Supplementary Data 1". Deposited raw quality controlled and assured CyTOF data (FCS files) can be accessed via a registered account without any subscriptions, through the Cytobank Premium platform (https://premium.cytobank.org/cytobank/projects/3668) via the following links. FCS files for Fig. 1 (https://premium.cytobank.org/cytobank/experiments/474341), Alum stimulation (https://premium.cytobank.org/cytobank/experiments/474346); MPLA stimulation (https://premium.cytobank.org/cytobank/experiments/474666); CpG stimulation (https://premium.cytobank.org/cytobank/experiments/474672) and R848 stimulation (https://premium.cytobank.org/cytobank/experiments/472809) related to Figs. 2–7. Participant's identifier with the FCS file names along with independent assay number are provided in the Supplementary Excel file named "Supplementary Data 2". The same Excel file is also available in the "Attachments" section (in the Cytobank platform) under each stimulation. For any further information, please email the corresponding author Dr. David Dowling at david.dowling@childrens.harvard.edu or our PVP CyTOF team at pvp.cytof@childrens.harvard.edu.

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

## Acknowledgements

The study was supported by Adjuvant Discovery Program (75N93019C00044) (to O.L. and D.J.D.), Adjuvant Development Program (272201800047C) (to O.L. and D.J.D.), and Development of Vaccines for the Treatment of Opioid Use Disorder (272201800047C-P00003-9999) (to O.L. and D.J.D.). The *Precision Vaccines Program* (PVP) is supported, in part, by the Boston Children's Hospital, Department of Pediatrics. The authors also thank the members of the PVP for helpful discussions, the PVP Data Management and Analysis Core (DMAC) including Dr. Joann Arce and Caitlin Syphurs for assistance with data QA, as well as Drs. Kevin Churchwell, Wendy Chung, Gary Fleisher, David Williams, and Nancy Andrews for their support of the PVP. S.B. thanks Eric Haas from Ionic Cytometry Solutions, Arsh Patel from Longwood Medical Area CyTOF Core at Dana-Farber Cancer Institute (DFCI), Megan Perkins and Vinicius Motta from Standard BioTools and Qianjun Zhang from Beckman Coulter for their technical assistance and feedback.

## Author contributions

Experiments and CyTOF acquisition were conducted by S.S.S. and J.D. CyTOF data analysis, algorithm implementation and presentation in figures were done by S.B. and D.J.D. with help from R.M.G. The manuscript was written by S.B. and D.J.D. with feedback from S.S.S., R.M.G., D.S., J.D., L.R.B. and O.L. O.L. and D.J.D. conceived the project, secured funding, and supervised the study.

## Competing interests

The authors declare the following competing financial interest(s): D.S., O.L. and D.J.D. are named inventors on patents relating to small molecule adjuvants assigned to Boston Children's Hospital. O.L.'s laboratory has received sponsored research support from *GlaxoSmithKline* (GSK) and O.L. has served as a consultant to GSK and *Hillevax*. D.J.D. is on the scientific advisory board of *EdJen BioTech* and serves as a consultant with Merck Research Laboratories/Merck Sharp & Dohme Corp. (a subsidiary of Merck & Co., Inc.). O.L. and D.J.D. are cofounders of *Ovax Inc*. These commercial or financial relationships are unrelated to the current study. R.M.G. is an Application Fields Scientist for the Cytobank platform at Beckman Coulter Life Sciences. The rest of the authors declare no competing interests.
