## [Peer Review File · Communications Biology]

Reviewers' comments:

Reviewer #1 (Remarks to the Author):

The authors propose a well-conducted study evaluating the immunophenotypic characterization of the human MNC compartment by age, with functional study on the effect of multiple PRRa in neonates, adults and elders through a multiparametric evaluation.

Despite intrinsic limitations, with particular concern on the use of cord blood for neonates and single time point evaluation not extended to host factors that are essential for immune functions, this study may be used as initial background to design adjuvanted vaccine formulations for age-tailored immunization.

Reviewer #2 (Remarks to the Author):

The manuscript by Schüller et al, "Immune profiling of age and adjuvant-specific activation of human blood mononuclear cells in vitro" describes the effect of multiple PRRa on MNCs activity by using a single time point stimulation in vitro system. It is widely known that vaccines responses in newborn and elder populations are far from be ideal, since ineffective protection is achieved in most cases. Therefore, current vaccines, as the majority of drugs, need crucial improvements. Authors proposed in this paper an age-tailored adjuvantation system for newborn and elders that ensure immune protection, based in the use of four different PRRa. Although the manuscript is very interesting and the assays are well defined and justified, there are major points that should be addressed by the authors before manuscript acceptance:

- Activation profile study of co-stimulatory molecule CD86 (figure 2) and CD40 (figure 3) is only showed in elder cohort, whereas newborn as well as adult' cohorts are missing. Authors should include, at least in supplementary material, these Med MIs (10 participants for newborn cohort and 10 participants for the adult). Further, authors did not explain why they present elder cohort's Med MI in Figures 2A and 3A, whereas others cohorts are missing.
- Figure 2: statistical analysis for newborn cohort both in mDCs and pDCs populations are missing. Authors should include this analysis in the figure in order to better interpretate the results.
- Figure 3 results showed that R848 adjuvant drives CD40 upregulation on DCs but also in monocytes. This fact should be emphasized in Figure 3' legend, since it only makes reference to DCs. Further, authors should explain why R848 has no significant effect on CD40 regulation in monocytes in adult cohort.
- Data showed in Figure 4B and Figures 5A, B and C, are exactly the same. The only difference is that, statistical analysis in not included in Figure 4B. In order to avoid duplication of the data, my recommendation is to delete Figure 4B. Figure 4A can be maintained or moved to Supplemental material.
- Figure S7: statistical significance for R848 adjuvant should be included in all plots, as is the one that has a more relevant effect. For instance, significant analysis was not included in Fig S7E (elder cohort) and it looks like R848 induces an increase in IL10+ cells.
- Line 229: authors declare that "such induction was visible over alum treated newborn MNCs but absent in adult and elder groups", however, no changes in TNF population was observed after alum

treatment. Please explain this point.

- Line 230: R848 triggered significant IL-23p29 responses over alum in newborn, however, this effect should be considered as minor, since comparison between R848 and control group is not statistically significant. The same observation is valid for CpG comparison with alum. Authors comparison, for both cases, is between two different adjuvants, and no with control cells (unstimulated), therefore, the relevance of these observations is clearly minor.
- Figure S7B: R848 effect on TNF+ cells is statistically significant compared with control cells (*), however, this significance is difficult to see and it is not according to the observed in Figure 5D. Authors should explain this point.
- Line 234: authors indicated that no differences were observed for IL-17 in the three different age groups. After observed Fig S7D, it is worth to note that IL17+ cell percentages are clearly higher for one of the participants in all conditions (control, Alum, MPLA, CpG and R848) (outlier). This need to be taken into account. It is also relevant that participants were excluded from the analysis because of low cell acquisition (<70K) without indicating CD45+ CD66a- MNCs acquired cells. In order to avoid participants' exclusion, cell limit might be considered lower than 70K. Consequently, extra participants could be included to perform the study and statistical analysis would be more precise and conclusive.
- Line 329: authors declare that R848 induced cytokines such as IL-23p19, TNF and IL-10. However, results obtained in Figure S7, are not in accordance with this affirmation. There is no statistical difference between R848 and control cells for the three cytokines previously mentioned. Authors state this effect based on alum response, lower than control, without focused on the fact that alum stimulated cells did not respond the same way as control cells. This clearly need a more depth discussion from authors.

Minor points:

- Custom conjugated antibodies' source is not clear, as in line 428 the authors indicated that they were obtained from the Harvard Medical Area CyTOF Antibody Resource and Core and in Supplementary Table 1 the indicated source is Miltenyi.
- Cell-ID Cisplatin-195Pt' as well Cell-ID Intercalators sources should be included in Supplementary Table 3, not 1. Further, authors did not mention the specific Cell-ID Intercalator used (191Ir or 193Ir) in their experiments.
- Line 525: D.D. abbreviation should be D.J.D.
- Figure Legends 2, 3. and 4: participants' number should be n=7-10, instead of 7-11, considering all the results come from the same inicial participants (already excluding 3 of them for low number of cells (<70K)).
- Mononuclear cells abbreviation should be consistent throughout the text (MNCs and not BMCs). Figure S4, S5, S6 & S7, lines 289, & 325.

Reviewer #3 (Remarks to the Author):

The manuscript "Immune profiling of age and adjuvant-specific activation of human blood mononuclear cells in vitro" by Schuller et al, describes and compares the immunostimulating effect of three PRR agonist different adjuvants and Alum on blood mononuclear cells in three cohorts of subjects, newborn, adult, and elderly. By applying a computational approach to the analysis of the mass cytometry multidimensional data, the authors demonstrate the immunostimulating properties of R484 in activating innate cells and promoting the IFN-g release in an age-independent way. This work is extremely

interesting considering the challenges of immune changes at the extreme of age, such as newborn and elderly, that represent, for most of the vaccines, the target populations. The study is well organized, presenting different immunological outputs (APC activation, intracellular cytokine production by both innate and adaptive cells) across the different cohorts upon stimulation of the respective cells with the selected adjuvants. Specific comments for the different sections are reported below:

-Introduction

The authors should introduce and describe here the two principal components of their analysis, i.e the impact of age on the immune system responsiveness, and the mode of action and commercial use of the selected adjuvants. To this aim I would suggest moving lines 298-316 from the discussion to the introduction.

The description of the CyTOF technology and high-dimensional data computational tools (lines 94-119) could be consistently shortened considering its well-known application in immunological field for a long time. Most of the text could be replaced by appropriate references.

-Results

In lines 189-193 the CpG activity is discussed only in comparison with other PRR agonist adjuvants, despite significant differences were observed also versus Alum in most of the conditions reported; please revise this point.

Lines 182-188 and 219-227 should be shortened in the results, and eventually moved to the discussion, when the modulation of the different cytokines will be discussed. Moreover, the reported list of functions could be reductive with respect to the multiple activity of these cytokines and to the multiple cell types involved in their release.

Concerning the analysis of Figure 7, have the authors investigated which are the phenotype differences among nodes that were significantly more abundance in adult gammadelta T cells (16, 22 and 30)?

-Discussion

The authors should better discuss the impact of R484 in the cohort of elderly people; indeed, the strong IFN-g induction should be discussed in the context of the chronic inflammatory status (inflammaging) associated with age, that should not be exacerbated by adjuvants.

In line 377, point (a) the authors should add also the limitation related the use of a single dose for each adjuvant.

Throughout the manuscript, the author should use a single acronym for blood cells, choosing between BMC or MNC.

Reviewer comments:

Reviewer #1

General comment: The authors propose a well-conducted study evaluating the immunophenotypic characterization of the human MNC compartment by age, with functional study on the effect of multiple PRRa in neonates, adults and elders through a multiparametric evaluation.

RESPONSE: The authors appreciate the reviewer's summary and positive comments. We have incorporated the suggestions from all three reviewers and hope the revised version of the manuscript is now acceptable for publication.

Comment (1): Despite intrinsic limitations, with particular concern on the use of cord blood for neonates and single time point evaluation not extended to host factors that are essential for immune functions, this study may be used as initial background to design adjuvanted vaccine formulations for age-tailored immunization.

RESPONSE: We agree with the reviewer's insight, we appreciate that the reviewer already identified the specific aim of this study. We have documented our study limitations including usage of cord blood and single time point evaluation of activation profiles of cell populations in the "Discussion" section (lines 393-399 in marked up version of the revised manuscript). For this study, we focused on hypothesis generation via an *in vitro* modelling approach and use such data to undertake complementary studies, to model responses to adjuvants and/or vaccines in early life (e.g., PMIDs 28352660 and 36258023) or in elders (e.g., PMIDs 34783582 and 36788219).

Reviewer #2

General comment: The manuscript by Schüller et al, “Immune profiling of age and adjuvant-specific activation of human blood mononuclear cells in vitro” describes the effect of multiple PRRa on MNCs activity by using a single time point stimulation in vitro system. It is widely known that vaccines responses in newborn and elder populations are far from be ideal, since ineffective protection is achieved in most cases. Therefore, current vaccines, as the majority of drugs, need crucial improvements. Authors proposed in this paper an age-tailored adjuvantation system for newborn and elders that ensure immune protection, based in the use of four different PRRa. Although the manuscript is very interesting and the assays are well defined and justified, there are major points that should be addressed by the authors before manuscript acceptance:

RESPONSE: The authors appreciate the reviewer’s summary and positive comments. We have incorporated the suggestions from all three reviewers and hope the revised version of the manuscript is now acceptable for publication.

Comment (1): Activation profile study of co-stimulatory molecule CD86 (figure 2) and CD40 (figure 3) is only showed in elder cohort, whereas newborn as well as adult’s cohorts are missing. Authors should include, at least in supplementary material, these Med MIs (10 participants for newborn cohort and 10 participants for the adult). Further, authors did not explain why they present elder cohort’s Med MI in Figures 2A and 3A, whereas others cohorts are missing.

RESPONSE: Below is helpful additional clarity on the points mentioned by the reviewer.

First, to help address the reviewer’s point, the re-submitted manuscript now contains newly generated Supplementary Fig. 4 and Supplementary Fig. 5, which portrays activation profiles of CD86 (Fig. S4) and CD40 (Fig. S5) on mDCs, pDCs and monocytes from all the participant’s (N=27) in a histogram format.

Second, given the extensive depth of data generated in our study, we selected one age group per figure to graphically represent an exemplar of each activation profile seen in mDCs (i.e., CD86 histogram in Figure 2A and CD40 histogram in Figure 3A). Since the annotated data for each age group is also provided in each figure (i.e., Figure 2B-D Figure 3B-D), along with statistical data, and given the journal’s limitation on manuscript size, it would be challenging and excessive to provide graphs for all instances. Of note, all the raw values and analyzed necessary fold change between groups of interest being included in the supportive Excel file named “Supporting Data 1” as a supplement.

ACTION: To enhance clarity and address the reviewer’s observation about “missing data”, the authors have modified the figure legends in Figure 2 and 3 which now as “(A) *Representative* median metal intensity (Med MI) of CD86 expression on mDCs in the elder cohort” in Figure 2 and “(A) *Representative* median metal intensity (Med MI) of CD40 expression on mDCs in the elder cohort” in Figure 3. Newly generated Supplementary Fig. S4 and S5 were included in revised version, where the histogram profile of CD86 and CD40 from all participants were displayed.

Comment (2): Figure 2: statistical analysis for newborn cohort both in mDCs and pDCs populations are missing. Authors should include this analysis in the figure in order to better interpretate the results.

RESPONSE: In the first submission, we did not add any statistical analysis intentionally as we did not find any statistical significance between groups.

ACTION: As per reviewer's suggestion, we added the statistical analysis in mDC and pDC profiles (Figure 2B & C) from newborn cohorts. Of note, all the raw values and analyzed necessary fold change between groups of interest being included in the supportive Excel file named "Supporting Data 1" as a supplement.

Comment (3): Figure 3 results showed that R848 adjuvant drives CD40 upregulation on DCs but also in monocytes. This fact should be emphasized in Figure 3's legend, since it only makes reference to DCs. Further, authors should explain why R848 has no significant effect on CD40 regulation in monocytes in adult cohort.

RESPONSE: We thank to the reviewer to notify the important point. We modified the Figure 3's legend and now it reads as "TLR7/8a R848 drives CD40 upregulation on DCs and monocytes".

For the adult cohort (Figure 3D) we did not find any statistically significant and therefore we measured the mean fold change of other PRRa's response compared with R848 and the fold changes are documented at the top of the statistical results. We found 4.12-fold CD40 upregulated in R848 stimulated adult monocytes when compared with control. Similar trends were found when compared with the alum treated group (16.07-fold), MPLA treated group (4.35-fold) and CpG treated group (1.44-fold). By this observation we can conclude that R848 has ability to upregulate CD40 in adult monocytes, but we did not get any statistically significant perhaps due to the cohort's nature-i.e. some participants are high responders whereas others are moderate to low responders in R848 stimulated groups.

ACTION: In response to the reviewer's comments, we added the statistical analysis for monocytes profiles from adult the cohort. We added the mean fold change over R848 (in adult and newborn monocytes) where we could not achieve the statistical significance. Of note, all the raw values and analyzed necessary fold change between groups of interest being included in the supportive Excel file named "Supporting Data 1" as a supplement.

Comment (4): Data showed in Figure 4B and Figures 5A, B and C, are exactly the same. The only difference is that, statistical analysis is not included in Figure 4B. In order to avoid duplication of the data, my recommendation is to delete Figure 4B. Figure 4A can be maintained or moved to Supplemental material.

RESPONSE: We appreciate the reviewer's suggestion, and we agreed about data repetition. The initial presentation was attempting to provide the data side-by-side for ease of understanding.

ACTION: As per reviewer's suggestion, we **deleted Figure 4B** and kept Figure 4A (now Figure 4) as a standalone figure.

Comment (5): Figure S7: statistical significance for R848 adjuvant should be included in all plots, as is the one that has a more relevant effect. For instance, significant analysis was not included in Fig S7E (elder cohort) and it looks like R848 induces an increase in IL10+ cells.

RESPONSE: Previously, we did not add any statistical analysis intentionally where we did not find any statistical significance between groups. In addition, we agree with the reviewer that R848 induces an increase in IL10⁺ cells (which we already mentioned in line 244 in marked up version of the revised manuscript).

ACTION: As per reviewer's suggestion, we added the statistical analysis for R848 in all plots (Fig. S9). We also added fold changes in Fig S7E (Fig. S9E in revised manuscript) for better understanding between groups where we did not find any statistical significance. Of note, all the raw values and analyzed necessary fold change between groups of interest being included in the supportive Excel file named "Supporting Data 1" as a supplement.

Comment (6): Line 229: authors declare that "such induction was visible over alum treated newborn MNCs but absent in adult and elder groups", however, no changes in TNF population was observed after alum treatment. Please explain this point.

RESPONSE: We think the reviewer might miss the point that in the newborn cohort (previously Fig. S7B, now Fig. S9B) R848 mediated TNF response (magenta colored group and asterisks were in magenta) is statistically significant when compared with alum treated group (blue group, **P < 0.01). In addition, we did not observe such statistically significant TNF upregulation (R848 versus alum) in adults as well as in the elder cohort.

Comment (7): Line 230: R848 triggered significant IL-23p29 responses over alum in newborn, however, this effect should be considered as minor, since comparison between R848 and control group is not statistically significant. The same observation is valid for CpG comparison with alum. Authors comparison, for both cases, is between two different adjuvants, and no with control cells (unstimulated), therefore, the relevance of these observations is clearly minor.

RESPONSE: We agree with the reviewer that in both cases, we did not find any statistical significance when compared with the control group.

First, for more clarification, we added fold changes in Fig. S9C between groups where we did not find any statistical significance. R848 mediated IL-23p19 response was statistically significant in newborn, whereas 1.63-fold and 1.78-fold in adult and elder cohorts respectively, when compared with alum treated groups. Therefore, R848 mediated IL-23p29 responses over alum in newborn are not probably minor.

Second, when compared with control groups, R848 mediated IL-23p29 responses were 1.51-fold higher in the newborn cohort, 1.26-fold in the adult cohort and 5.66-fold in elder. Despite elevated levels of fold changes, we did not find any statistical significance, perhaps because of the presence of a high responder participant in the control group of newborn and adult cohort. Taking into consideration the fact that it is human biology, therefore we have not modified the line 230 (line 238 in marked up version of the revised manuscript).

Third, to be consistent throughout the manuscript, we have also added the mean-fold changes in Fig. S7B (Fig. S9B in the revised manuscript) between two groups - 1) CpG versus control and 2) CpG versus alum in all the cohorts for better clarification. CpG mediated TNF response was 2.69-fold higher (which is statistically significant) when compared with alum in the newborn cohort, 1.49-fold in the adult cohort and 1.30-fold in elder.

ACTION: To improve clarity, the revised manuscript now includes statistical analysis for R848 and CpG in all plots, such as Fig. S9B, as well as in Fig. S9C. In addition, we have now documented the fold change in Fig. S9 between groups of interest to support our observation. Of note, all the raw values and analyzed fold changes being included in the supportive Excel file named “Supporting Data 1” as a supplement.

Comment (8): Figure S7B: R848 effect on TNF+ cells is statistically significant compared with control cells (*), however, this significance is difficult to see and it is not according to the observed in Figure 5D. Authors should explain this point.

RESPONSE: Figure S7B (Fig. S9B in revised manuscript) is representing % of MNCs expressing TNF. A significant response is observed in Fig. S9B (newborn), comparing R848 treatment against the control, alum and MPLA. Fig 5D in contrast is presented as a mean fold change of over unstimulated cells, which has the visual benefit of being easier to interpret but reduces the dimensionality of the data to not take into account outliers. As such, we have historically provided both analysis in this and prior papers (see PMIDs 28343701 and 34783582 as examples).

Comment (9): Line 234: authors indicated that no differences were observed for IL-17 in the three different age groups. After observed Fig S7D, it is worth to note that IL17+ cell percentages are clearly higher for one of the participants in all conditions (control, Alum, MPLA, CpG and R848) (outlier). This need to be taken into account. It is also relevant that participants were excluded from the analysis because of low cell acquisition (<70K) without indicating CD45+ CD66a- MNCs acquired cells. In order to avoid participants’ exclusion, cell limit might be considered lower than 70K. Consequently, extra participants could be included to perform the study and statistical analysis would be more precise and conclusive.

RESPONSE: We appreciate the reviewer’s observation. We agree with the reviewer that one of the participants could be an outlier as a high responder. Also, IL-17 production is canonically associated with antigen specific anamnestic recall responses in Th17 polarized CD4⁺ T cell. Therefore, we did not expect to see, much IL-17 responses in these panels.

Regarding high and low responders, it is debatable to include all the participants even if there are less than 70K live MNCs (live CD45⁺ CD66a⁻ cells). As IL-17⁺ cells are rare populations, if the live MNCs will be less than 70K, then we think the quality of the data could be jeopardized. As a good lab practice, whenever we deal with rare populations, we try to maintain a threshold of live cells (or live MNCs), which was for to 70K (during analysis) for this study, to be a) more confident about data quality, b) to allow for the capturing the quantity of the rare populations and c) to maintain the excellent quality of HD reduction topographical images.

Comment (10): Line 329: authors declare that R848 induced cytokines such as IL-23p19, TNF and IL-10. However, results obtained in Figure S7, are not in accordance with this

affirmation. There is no statistical difference between R848 and control cells for the three cytokines previously mentioned. Authors state this effect based on alum response, lower than control, without focused on the fact that alum stimulated cells did not respond the same way as control cells. This clearly need a more depth discussion from authors.

RESPONSE: We appreciate the reviewer's concerns. As we have already outlined in the response to comment 7, we did not find any statistical significance when compared with the control group. As outline in Comment 7, and to be consistent throughout the manuscript, we added fold changes of R848 mediated cytokines (IL-23p19, TNF and IL-10) responses compared with targeted PRRa as well as compared with control. The fold changes of R848 mediated cytokine responses (IL-23p19, TNF and IL-10) vary between 1.51 to 5.66 when compared with control (Fig S9B, C & E).

ACTION: To improve clarity, the revised manuscript now includes statistical analysis for R848 and CpG in all plots, such as Fig. S9B, as well as in Fig. S9C. In addition, we have now documented the fold change between groups of interest to support our observation which is documented in line 245 (R848 mediated IL-23p19 response); TNF induction in Fig S9B (adult and elder cohorts); IL-23p19 induction in Fig S9C (all cohorts) and lastly IL-10 induction in Fig S9E (all cohorts). Of note, all the raw values and analyzed necessary fold change between groups of interest being included in the supportive Excel file named "Supporting Data 1" as a supplement.

Minor points:

Comment (11): Custom conjugated antibodies' source is not clear, as in line 428 the authors indicated that they were obtained from the Harvard Medical Area CyTOF Antibody Resource and Core and in Supplementary Table 1 the indicated source is Miltenyi.

RESPONSE: We apologize for the misunderstanding. We purchased the pure unconjugated antibodies for CyTOF either from the vendor Miltenyi (IFN α and CD80) or BioLegend (TCR $\gamma\delta$) according to the Supplementary Table 1. The custom conjugation of the purchased antibodies with desired metal isotope were accomplished by the Harvard Medical Area CyTOF Antibody Resource and Core. For better understanding of this method, line 428 (line 444 in marked up version of the revised manuscript) now read as "*Custom conjugation of certain antibodies (specifically, IFN α , CD80 and TCR $\gamma\delta$) were accomplished at Harvard Medical Area CyTOF Antibody Resource and Core (Boston, MA) according to the Core's optimized protocol*".

Comment (12): Cell-ID Cisplatin-195Pt' as well Cell-ID Intercalators sources should be included in Supplementary Table 3, not 1. Further, authors did not mention the specific Cell-ID Intercalator used (191Ir or 193Ir) in their experiments.

RESPONSE: We appreciate the reviewer's suggestion.

First, Cell-ID Cisplatin-195Pt is used to eliminate dead cells and Cell-ID Intercalator (191Ir and 193Ir) were used to identify nucleated cells. We did not mention it separately as this is seen as a standard operating procedure during a CyTOF based stain. In Fig S2, live cells are gated as 195Pt⁻ and nucleated cells are gated as 191Ir⁺ followed by 193Ir⁺ cells. As per reviewer's suggestion we have updated these details in line 456 (line 474 in marked up version of the revised

manuscript), which now read as “*and gating live (195Pt) nucleated (191Ir⁺ and 193Ir⁺) cell events (Supplementary Fig. 2)*”.

Second, we did not include those metal tagged reagents in Supplementary Table 3 as these (195Pt,191Ir and 193Ir) are the part of the immunophenotyping panel not the secondary reagents.

Comment (13): Line 525: D.D. abbreviation should be D.J.D.

RESPONSE: We apologize for the error. We corrected it.

Comment (14): Figure Legends 2, 3. and 4: participants’ number should be n=7-10, instead of 7-11, considering all the results come from the same inicial participants (already excluding 3 of them for low number of cells (<70K)).

RESPONSE: For the clarification, the number of participants is ranging from 7-11. The lowest number of participants (n=7) were in the control elder cohort and highest number (n=11) was in the adult cohort stimulated with CpG. The Supplementary Excel file named “Supplementary note 1”, contains a table with this exact information.

Comment (15): Mononuclear cells abbreviation should be consistent throughout the text (MNCs and not BMCs). Figure S4, S5, S6 & S7, lines 289, & 325.

RESPONSE: We appreciate the feedback. These terms were chosen to clearly define three related, but distinctly processed primary human cells, 1) PBMCs, 2) MNCs, and 3) BMCs. First, the term “PBMCs” or “CBMCs” (peripheral or cord blood mononuclear cells) are usually used to identify cells isolated by Ficoll treatment. Secondly, during the immunophenotyping analysis of PBMCs or CBMCs, we gated out the granulocyte population (CD45⁺ CD66a⁺) and we termed the resultant cells (CD45⁺ CD66a⁻) as “Mononuclear cells” (MNCs). Thirdly, the term “BMC”, which stands for “Blood mononuclear cells”, was used in the manuscript to help distinguish between instances where we are using peripheral or cord blood, as opposed to the use of any isolated mononuclear cells (PBMCs or MNCs). As our team has standardized these terms across this and prior studies, we prefer to maintain the use outlined in the first submission.

Reviewer #3

General comment: This work is extremely interesting considering the challenges of immune changes at the extreme of age, such as newborn and elderly, that represent, for most of the vaccines, the target populations. The study is well organized, presenting different immunological outputs (APC activation, intracellular cytokine production by both innate and adaptive cells) across the different cohorts upon stimulation of the respective cells with the selected adjuvants.

RESPONSE: The authors appreciate the reviewer's summary and positive comments. We have incorporated the suggestions from the 3 assigned reviewers and hope the revised version of the manuscript is now acceptable for publication.

Comment (1): Introduction: The authors should introduce and describe here the two principal components of their analysis, i.e. the impact of age on the immune system responsiveness, and the mode of action and commercial use of the selected adjuvants. To this aim I would suggest moving lines 298-316 from the discussion to the introduction.

RESPONSE: We agree. We have now moved lines 298-316 from the "Discussion" section to the "Introduction" section (now it's line 119-137).

Comment (2): Introduction: The description of the CyTOF technology and high-dimensional data computational tools (lines 94-119) could be consistently shortened considering its well-known application in immunological field for a long time. Most of the text could be replaced by appropriate references.

RESPONSE: We appreciate the reviewer's suggestion. While also considering the feedback from Reviewers #1 and #2, we have shortened this section to the best of our ability.

Comment (3): Results: In lines 189-193 the CpG activity is discussed only in comparison with other PRR agonist adjuvants, despite significant differences were observed also versus Alum in most of the conditions reported; please revise this point.

RESPONSE: We appreciate the reviewer's suggestion. This has been rectified in marked up version of the revised manuscript (lines 197-201), which now reads as "*Newborn and adult CD11c⁺ monocytes showed superiority in CD86 induction upon CpG stimulation as compared to R848 and alum (Fig. 2D & Supplementary Fig. 4). The same trend was also portrayed on the CD11c⁺ CD14⁺ mDCs in adults and elders but not in newborn mDCs (Fig. 2B) or in CD11c⁺ CD123⁺ pDCs from newborn, adult and elder cohorts only when compared with alum stimulated groups (Fig. 2C)*".

Comment (4): Results: Lines 182-188 and 219-227 should be shortened in the results, and eventually moved to the discussion, when the modulation of the different cytokines will be discussed. Moreover, the reported list of functions could be reductive with respect to the multiple activity of these cytokines and to the multiple cell types involved in their release.

RESPONSE: We appreciate the reviewer's suggestion. Before explaining the results, we briefly re-introduced the salient activation markers (now lines 189-195) and cytokines (now lines 227-234) for the benefit of readers who are not from a core immunophenotyping background. We hope the reviewer supports this editorial approach.

Comment (5): Results: Concerning the analysis of Figure 7, have the authors investigated which are the phenotype differences among nodes that were significantly more abundance in adult gamma delta T cells (16, 22 and 30)?

RESPONSE: Thank you for the excellent suggestion. We have now further analyzed nodes 16, 22 and 30 for the expression of TCRgd (as a positive marker because those cells are gamma delta T cells) along with memory phenotype markers CD45RA, CD27 and CCR7. We also investigated cytotoxicity marker CD56.

ACTION: The generated data is presented as a histogram and newly generated graph format in Supplementary Figure 13 (Fig. S13). We also included the observation in lines 290-301 and now read as *"We further immunophenotyped nodes 16, 22 and 30 according to the expression of memory phenotype defining markers CD45RA, CD27 and CCR7 (PMID: 32830910 and 29867961) along with $\gamma\delta$ T cells marker TCRgd. As speculated, TCRgd expression is high throughout the nodes. Low expression of CD45RA and variable (low to high) expression of CD27 defined a mixed central memory ($CD45RA^{low} CD27^{high}$), and effector memory ($CD45RA^{low} CD27^{low}$) phenotypes among nodes 16, 22 and 30. Conclusively, CCR7 upregulation in node 30 defined a central memory ($CD45RA^{low} CCR7^{high}$) phenotype whereas nodes 16 and 22 were consisted of effector memory ($CD45RA^{low} CCR7^{low}$) $\gamma\delta$ T cells. The abundance of cytotoxic $CD56^+$ $\gamma\delta$ T cells were observed in node 16 when compared with node 30. Cells in node 16 which are $IFN\gamma$ producing $CD56^+$ effector $\gamma\delta$ T cells might possess cytotoxic activity classically associated with anti-tumor responses (PMID: 25960933 and 33603738)"*.

Furthermore, we included additional description in "Methods" section (lines 499-502) and now read as *"Next, we exported nodes 16, 22 and 30 from the SPADE bubble of R848 stimulated adult $IFN\gamma^+$ $\gamma\delta$ T cells using the Cytobank platform and portrayed the phenotypic differences based on memory signature (CD45RA, CD27 and CCR7) along with functional (CD56) profile"*.

Comment (6): Discussion: The authors should better discuss the impact of R484 in the cohort of elderly people; indeed, the strong IFN-g induction should be discussed in the context of the chronic inflammatory status (inflammaging) associated with age, that should not be exacerbated by adjuvants.

RESPONSE: Despite the hypothetical concerns of the chronic inflammatory status (inflammaging) associated with age, 'HEPLISAV-B' and 'CORBEVAX' both ODN CpG adjuvanted vaccines, and 'COVAXIN', a Alhydroxiqum-II (TLR7/8) adjuvanted vaccine, have been administered to >150 million individuals worldwide, mostly in older adult. Phase 3 epidemiological data of 'HEPLISAV-B' (PMID: 36269938), 'CORBEVAX' (PMID: 37113012) and 'COVAXIN' (PMID: 35681012) does not linking any evidence of phenomena associates with inflammaging or reactogenicity. The latest report of the active surveillance study also did not reveal any adverse events associated with inflammaging or reactogenicity (PMID: 37641637).

ACTION: To better address the reviewers concern, we added the above description in the revised manuscript (lines 334-339), which now reads as “Furthermore, experience with several adjuvanted vaccines employing TLR agonists, including COVAXIN, a Alhydroxiqum-II (TLR7/8a) adjuvanted vaccine as well as HEPLISAV-B and CORBEVAX both ODN CpG (TLR9a) adjuvanted vaccines, has been encouraging. These vaccines have been administered to >150 million individuals worldwide, many of whom were older adults, with safety surveillance data suggesting general safety and any phenomena that could potentially be attributed to exacerbation of inflammaging are rare”.

Comment (7): Discussion: In line377, point (a) the authors should add also the limitation related the use of a single dose for each adjuvant.

RESPONSE: We agree with the reviewer and line 377 (line 394 in marked up version of the revised manuscript), point (a) is now read as “(a) our study characterized MNCs at a single time point *with single concentration for selected adjuvants*, providing a snapshot of cellular composition and function”. The concentrations used have been optimized in multiple prior studies (e.g., PMIDs 22521247, 26274907, 28008331, 26933193, 27081760, 28352660, 28343701, 29312305 and 34783582).

Comment (8): Throughout the manuscript, the author should use a single acronym for blood cells, choosing between BMC or MNC.

RESPONSE: We appreciate the feedback. These terms were chosen to clearly define three related, but distinctly processed primary human cells, 1) PBMCs, 2) MNCs, and 3) BMCs. First, the term “PBMCs” or “CBMCs” (peripheral or cord blood mononuclear cells) are usually used to identify cells isolated by Ficoll treatment. Secondly, during the immunophenotyping analysis of PBMCs or CBMCs, we gated out the granulocyte population (CD45⁺ CD66a⁺) and we termed the resultant cells (CD45⁺ CD66a⁻) as “Mononuclear cells” (MNCs). Thirdly, the term “BMC”, which stands for “Blood mononuclear cells”, was used in the manuscript to help distinguish between instances where we are using peripheral or cord blood, as opposed to the use of any isolated mononuclear cells (PBMCs or MNCs). As our team has standardized these terms across this and prior studies, we prefer to maintain the use outlined in the first submission.

REVIEWERS' COMMENTS:

Reviewer #2 (Remarks to the Author):

Author's second version of the manuscript has been reviewed. My comments (major and minor) were properly revised and changes were accordingly performed. The manuscript has now my approval to be published.

Reviewer #3 (Remarks to the Author):

The authors successfully addressed all my previous suggestions, therefore the article should be accepted for publication.